# NON-AUTOREGRESSIVE MACHINE TRANSLATION AS CONSTRAINED HMM

## ABSTRACT

In non-autoregressive translation (NAT), directed acyclic Transformers (DAT) (Huang et al., 2022c) have demonstrated their ability to achieve comparable performance to the autoregressive Transformers. In this paper, we first show that DAT is essentially a fully connected left-to-right Hidden Markov Model (HMM) (Baum et al., 1970), with the source and target sequences being observations and the token positions being latent states. Even though generative models like HMM do not suffer from label bias (Lafferty et al., 2001) in traditional task settings (e.g., sequence labeling), we argue here that the left-to-right HMM in NAT may still encounter this issue due to the missing observations at the inference stage. To combat label bias, we propose two constrained HMMs: 1) Adaptive Window HMM, which explicitly balances the number of outgoing transitions at different states; 2) Bi-directional HMM, i.e., a combination of left-to-right and right-to-left HMMs, whose uni-directional components can implicitly regularize each other's biases via shared parameters. Experimental results on WMT'14 En↔De and WMT'17 Zh↔En demonstrate that our methods can achieve better or comparable performance to the original DAT using various decoding methods. We also demonstrate that our methods effectively reduce the impact of label bias.[1]

## 1 INTRODUCTION

Autoregressive (AR) models (Hochreiter & Schmidhuber, 1997; Vaswani et al., 2017) have played a major role in various text generation tasks such as machine translation (Bahdanau et al., 2015; Luong et al., 2015) and text summarization (Rush et al., 2015). Despite being powerful, AR models still have some drawbacks: 1) slow inference speed due to the sequential generation property; 2) label bias (Bottou & Fogelman-Soulié, 1991; Lafferty et al., 2001; Goyal, 2022) due to the local normalization (e.g., softmax operation over the vocabulary in language modeling). To accelerate inference, non-autoregressive (NAR) models (Gu et al., 2018) were proposed as an alternative to AR models, and they can generate all the target tokens simultaneously by assuming that the target tokens are conditionally independent given the source sequence, i.e., $P(Y|X) = \prod_i P(y_i|X)$. Moreover, due to this independence assumption, they do not suffer from *label bias* even with local normalization.[2]

Nevertheless, conventional NAR models cannot rival the generation quality of AR models without knowledge distillation (Kim & Rush, 2016), which bypasses the *multi-modality*[3] problem in the training data (Gu et al., 2018; Zhou et al., 2020). Recently, directed acyclic Transformers (DAT) (Huang et al., 2022c) has largely resolved the multi-modality issue by constructing a large directed acyclic graph on top of the decoder outputs and modeling one possible translation as a pathway in the graph. In Section 3, we show that DAT is essentially a fully connected *left-to-right HMM* (see Fig. 1a) (Nilsson & Ejnarsson, 2002), with the source and target sequences $(X, Y)$ being observations and the token positions in the graph being states. Then, we argue that the application of left-to-right HMM in NAT can bring back the label bias problem due to the special decoding objective, that is, to find the target observation $Y$ given the source observation $X$. This is different from the traditional application of HMM, e.g., sequence labeling, where the decoding objective is to find the

---

[1]Code is available in the supplementary materials.

[2]Here we focus on fully NAR models where the inference can be done in a single forward pass.

[3]The target translation distributions can be highly multi-modal (Gu et al., 2018).

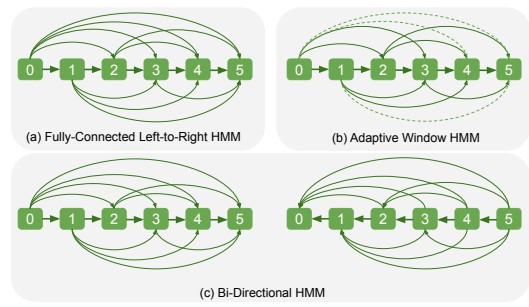

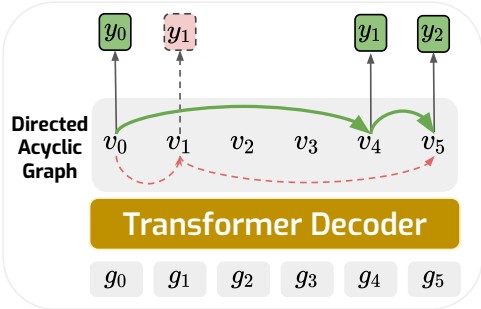

Figure 1: Graphical representations of different HMM models. The 6 numbers denote 6 states. Dashed lines in (b) are the removed paths.

Figure 2: Directed Acyclic Transformer. The encoder part is the same as standard Transformer so we omit it here.

latent states given all observations. The issue will bias the model towards choosing the decoding paths which contain states with less outgoing transitions, or equivalently, states with low-entropy transition probabilities (Goyal, 2022) (see Section 3).

To fix this issue, we propose two novel constrained HMMs: *Adaptive Window* HMM (AW-HMM) and *Bi-directional* HMM (Bi-HMM). For AW-HMM, the model will equalize the number of outgoing transitions for part of the latent states by dynamically determining the maximal number of future states that can be observed by the current state according to the source length $|X|$. For Bi-HMM, we combine a left-to-right and a right-to-left HMM, and allow them to share part of their parameters. Since the two HMMs would exhibit label bias issue from opposite directions, we hope that by sharing parameters, their respective biases can be canceled out to a certain extent. Experimental results on WMT'14 En↔De and WMT'17 En↔Zh show that our methods consistently achieve better or comparable performance to the original left-to-right HMM (i.e., DAT).

To summarize, our contributions are as follows:

- We establish the understanding that DAT is a fully connected left-to-right HMM, and argue it may suffer from label bias due to the missing observations during inference (Section 3).
- We propose AW-HMM, which adaptively balances the number of outgoing transitions at different latent states according to the source length, to combat label bias. We also propose Bi-HMM, whose uni-direction components can regularize each other's label bias implicitly (Section 4).
- Through experiments we show that the influence of label bias can be reduced using our methods (Section 6).

## 2 PRELIMINARIES

### 2.1 NON-AUTOREGRESSIVE MACHINE TRANSLATION

Compared to autoregressive machine translation, which generates the target tokens sequentially, $P(Y|X) = \prod_{i=0}^{m} P(y_i|y_{<i}, X)$ where $m$ is the target length, NAR models generate all the tokens simultaneously, i.e., $P(Y|X) = \prod_{i=0}^{m} P(y_i|X)$, and thus the inference would be faster. Besides, the inference of conventional NAR models is an independent token-level classification problem at each position, i.e., $\hat{y}_i = \arg\max_{\tilde{y}_i} P(\tilde{y}_i|X)$, and hence the local normalization $\sum_{y_i} P(y_i|X) = 1$ would not cause label bias (Section 3). Moreover, various training techniques (Kim & Rush, 2016; Gu et al., 2018; Qian et al., 2021; Gu & Kong, 2021) have been proposed to improve the performance (Section 7).

### 2.2 DIRECTED ACYCLIC TRANSFORMERS (DAT)

Recently, DAT (Huang et al., 2022c) (see Fig. 2) has narrowed the gap with AR models on translation tasks without knowledge distillation (Kim & Rush, 2016). DAT has the same architecture

as the standard Transformer except that it removes the causal attention mask in the decoder self-attention. The major difference is how it models $P(Y|X), Y = [y_0, y_1, ..., y_{m-1}]^\top \in \mathbb{R}^{m \times D}, y_0 = \texttt{BOS}, y_{m-1} = \texttt{EOS}$, and $D$ is hidden size.

DAT takes as input a source sentence $X = [x_0, x_1, ..., x_{n-1}]^\top \in \mathbb{R}^{n \times D}$ (to the encoder) as well as a sequence of learnable positional embeddings, $G = [g_0, g_1, ..., g_{L-1}] \in \mathbb{R}^{L \times D}$ (to the decoder), where $L$ is the decoder input length and $L = \lambda n > m, \lambda \in \mathbb{R}$. Then, it generates the output embeddings in the last decoder layer, $V = [v_0, v_1, ..., v_{L-1}]^\top \in \mathbb{R}^{L \times D}$. Next, it treats $v_i$ as vertices and the transitions $v_i \rightarrow v_j (j > i)$ as edges, forming a directed acyclic graph (see Fig. 2). Each path $A = [a_0, a_1, ..., a_{m-1}]$ from $a_0 = 0$ to $a_{m-1} = L - 1$ in the graph can yield the target translation $Y$, where $0 \leq a_i \leq L - 1$ is the position index that corresponds to the token $y_i$ and vertex $v_{a_i}$ in the graph. The path always starts with $a_0 = 0$ (i.e., $\texttt{BOS}$) and ends with $a_{m-1} = L - 1$ (i.e., $\texttt{EOS}$). Since DAT only allows transitions from small indices to large indices (i.e., left-to-right), we then have $a_i < a_j$ if $i < j$. The final training objective is:

$$P(Y|X) = \sum_{A \in \Gamma} P(Y, A|X) = \sum_{A \in \Gamma} P(Y|A, X)P(A|X) = \sum_{A \in \Gamma} \prod_{i=0}^{m-1} P(y_i|a_i, X)P(a_i|a_{i-1}, X)$$
(1)

where $\Gamma$ is the set of all possible paths $A$ with $|A| = |Y|$. $P(y_i|a_i, X)$ is the probability of predicting $y_i$ at the position $a_i$ while $P(a_i|a_{i-1}, X)$ is the probability of moving from position $a_{i-1}$ to $a_i$. To model $P(y_i|a_i, X)$, we do as follows:

$$P(y_i|a_i, X) = E[a_i, y_i], \qquad E = \textsc{Softmax}\left(V \cdot W_E\right) \in \mathbb{R}^{L \times \text{vocab}}$$
(2)

where $W_E \in \mathbb{R}^{D \times \text{vocab}}$. Similarly, $P(a_i|a_{i-1}, X)$ can be modeled as:

$$P(a_i|a_{i-1}, X) = T[a_{i-1}, a_i], \qquad T = \textsc{Softmax}\left(\frac{VW_K \cdot (VW_Q)^\top}{\sqrt{D}}\right) \in \mathbb{R}^{L \times L}$$
(3)

where $W_K, W_Q \in \mathbb{R}^{D \times D}$ are two linear transformations. Since we only allow left-to-right transitions, we set $T[i, j] = 0$, if $i \geq j$ (see the transition matrix in Fig. 3a).

At the inference stage, we can apply greedy or lookahead algorithm (Huang et al., 2022c) to solve $\hat{Y}, \hat{A} = \arg\max_{\tilde{Y}, \tilde{A}} P(\tilde{Y}, \tilde{A}|X)$ approximately or apply Viterbi algorithm (Shao et al., 2022) to obtain an exact solution. Another choice is to solve the marginal MAP problem $\hat{Y} = \arg\max_{\tilde{Y}} \sum_A P(\tilde{Y}, A|X)$ approximately by applying beam search (Huang et al., 2022c). Typically, the beam search performance is better than Viterbi/greedy/lookahead algorithms since $\arg\max_{\tilde{Y}} P(\tilde{Y}|X) = \arg\max_{\tilde{Y}} \sum_A P(\tilde{Y}, A|X)$ is the exact optimization problem of interest.

## 3 THE MISSING OBSERVATION AND LABEL BIAS

In this section, we argue that DAT is essentially a fully connected left-to-right HMM, and it can still suffer from the label bias problem at the inference stage due to the missing observation $Y$.

**DAT as A Fully Connected Left-to-Right HMM** If we compare the training objective Eq.1 of DAT to an HMM model, $(X, Y)$ can be seen as the observations and the positions $a_i \in [0, L - 1]$ as latent states in HMM, respectively. Specifically, $P(y_i|a_i, X)$ is the emission score, and $P(a_i|a_{i-1}, X)$ is the transition score, both of which are parameterized by a Transformer network. A normal HMM would allow transitions between any two states, while DAT only allows transitions from small positions to large positions ($P(a_i|a_{i-1}, X) = 0$, if $a_{i-1} \geq a_i$). Hence, it is essentially a fully connected left-to-right HMM or Bakis HMM (Nilsson & Ejnarsson, 2002) (see Fig. 1a).

However, there are some special properties when applying HMM to the translation task as opposed to common structured prediction problems such as sequence labeling. First, the observations are composed of two parts, i.e., source sentence $X$ and target sentence $Y$, and $X$ is always available whereas $Y$ is unavailable during inference. In sequence labeling, however, the observations only consist of $X$ (i.e., the text sequences to be labeled) and they are always given. Second, the decoding problem in machine translation is $Y = \arg\max_{\tilde{Y}} \sum_A P(\tilde{Y}, A|X)$ whereas in sequence labeling it is $A = \arg\max_{\tilde{A}} P(\tilde{A}|X)$. That is, we are interested in the missing "observation" $Y$ instead of the latent states $A$.

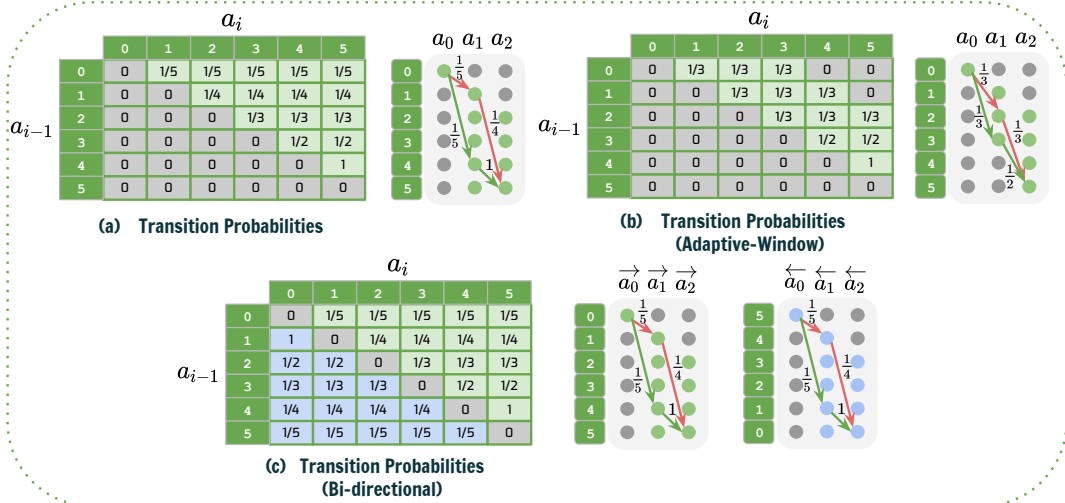

Figure 3: The transition probability matrices and two example decoding paths (in green and red) of DAT and our AW-HMM as well as Bi-HMM. We merge the left-to-right and right-to-left transition matrices in Bi-HMM (c) for simplicity. Grey circles denote that they are inaccessible according to the transition matrix.

**Label Bias**  The label bias problem refers to that when applying decoding algorithms that involve the comparison between cumulative scores (e.g., log probabilities) to a locally normalized discriminative model (e.g., Maximum Entropy Markov Model, McCallum et al., 2000), the states with less outgoing transitions (or low-entropy transition probability distribution) will be preferred over those with more due to the local normalization or "conservation of score mass" (Lafferty et al., 2001; Hannun, 2020; Goyal, 2022). Take the transitions $P(A|X) = \prod_i P(a_i|a_{i-1}, X)$ (Eq. 1) for example, which alone can be regarded as a locally normalized discriminative model. If each $P(a_i|a_{i-1}, X)$ follows a uniform distribution, then Viterbi decoding will pick the path that contains states with fewer outgoing transitions.

We give a toy example in Figure 3a where the Viterbi decoding result of $P(A|X)$ yields the green path with a probability $1/5$ rather than the red path with a probability $1/20$. The green path $(0, 4, 5)$ is preferred over the red path $(0, 1, 5)$ since $a_1 = 4$ has only one outgoing transition while $a_1 = 1$ has 4 outgoing transitions. In real scenarios, if the model is always biased to choose similar paths for different $X$ (i.e., being biased to put the tokens towards the end positions), the front positions can be wasted and the model's expressive power can be "under-utilized".

In HMM, however, we still have an emission term $P(y_i|a_i, X)$, which can freely "amplify" or "dampen" the influence of $P(a_i|a_{i-1}, X)$ based on the given observation $y_i$ (Lafferty et al., 2001). That is, $P(y_i|a_i, X)$ is able to upweight or downweight $P(a_i|a_{i-1}, X)$. In the toy example, if we consider the emission term $P(y_1|a_1 = 4, X)$ together with $P(a_1 = 4|a_0 = 0, X)$, the probability of choosing green path will be $P(y_1|a_1 = 4, X)/5$, and for the red path it is $P(y_1|a_1 = 1, X)/20$ (we ignore $P(y_0|a_0 = 0, X)$ and $P(y_2|a_2 = 5, X)$ since they are the same for both paths). Thus, there is still a chance that $P(y_1|a_1 = 1, X)/20 > P(y_1|a_1 = 4, X)/5$.

Unfortunately, ground truth $y_i$ is only provided during the training stage, not during the inference stage, and thus we do not know the ground truth emission score $P(y_i|a_i, X)$. We can only estimate $y_i$ through $\hat{y}_i$, $\hat{y}_i = \arg\max_{\tilde{y}_i} P(\tilde{y}_i|a_i, X)$ (see Eq. 2). Nevertheless, $\hat{y}_i$ would be dependent on $a_i$ in this case, whereas $y_i$ is not influenced by the choice of $0 \leq a_i \leq L - 1$. On one hand, the dependence between $a_i$ and $\hat{y}_i$ renders the emission term $P(\hat{y}_i|a_i, X)$ less effective for combating label bias. On the other hand, the estimation of $\hat{y}_i = \arg\max_{\tilde{y}_i} P(\tilde{y}_i|a_i, X)$ can now be adversely affected by the label bias problem in $P(a_i|a_{i-1}, X)$, which is directly related to the translation quality. This implies that we should not rely too much on the emission score for adjustment, and ideally the transition term can have less label bias by itself.

## 4 APPROACH

**Adaptive Window HMM (AW-HMM)**   We notice that one of the causes of label bias is that the number of the outgoing transitions (outdegree) is imbalanced at different positions (see Figure 1a, Figure 3a). At position 0, the number of possible outgoing transitions is $L-1$, whereas this number decreases to 1 at position $L-2$. To reduce this imbalance, we set the maximal number of outgoing transitions to be:

$$L' = \beta n, \beta < \lambda \in \mathbb{R}, \tag{4}$$

where $n$ is the source length. We call this an adaptive window since $L'$ changes according to the source length $n$. For positions $j < (\lambda - \beta)n$, they would have an equal number of outgoing transitions. Even though for positions $j > (\lambda - \beta)n$ the imbalance still exists, this adaptive window can still help alleviate the label bias issue in general. For example, we have $L' = 3$ in Figure 1b and 3b, and positions $0, 1, 2$ can all observe $L' = 3$ future positions, respectively. In Figure 3b, it can be seen that now the probability difference between the green and red paths has been narrowed from $3/20 = 1/5 - 1/20$ down to $1/18 = 1/6 - 1/9$. At first glance, we can choose a small $\beta$ to deal with label bias more effectively, but a small $\beta$ will lead to a small set $\Gamma$, which impairs the model's expressiveness. Therefore, we need to make a trade-off here (as we will see in Section 5.2). Note that the decoding algorithms of AW-HMM stay the same as DAT.

**Bi-directional HMM**   Instead of only modeling a left-to-right HMM (L2R), we consider both left-to-right and right-to-left HMM (R2L) simultaneously (see Figure 1c and 3c). The original objective function can be written as:

$$P(Y|X) = \alpha P(Y|X) + (1-\alpha)P(Y|X) = \alpha \sum_{\overrightarrow{A}} P(Y, \overrightarrow{A}|X) + (1-\alpha) \sum_{\overleftarrow{A}} P(Y, \overleftarrow{A}|X) \tag{5}$$

$\overrightarrow{A} = [\overrightarrow{a_0}, ..., \overrightarrow{a_{m-1}}]$ is the original latent path that goes from index $\overrightarrow{a_0} = 0$ to index $\overrightarrow{a_{m-1}} = L - 1$. $\overleftarrow{A} = [\overleftarrow{a_0}, ..., \overleftarrow{a_{m-1}}]$ is the opposite path that starts from $\overleftarrow{a_0} = L - 1$ and ends with $\overleftarrow{a_{m-1}} = 0$. Here, $\overrightarrow{a_i}$ corresponds to the token $y_i$ and vertex $v_{\overrightarrow{a_i}}$ while $\overleftarrow{a_i}$ corresponds to $y_{m-1-i}$ and $v_{\overleftarrow{a_i}}$. We use different transition parameters $[\overrightarrow{W_K}, \overrightarrow{W_Q}], [\overleftarrow{W_K}, \overleftarrow{W_Q}]$ (see Eq. 3) to model $P(\overrightarrow{A}|X)$ and $P(\overleftarrow{A}|X)$ but allow them to share Transformer backbone and emission parameters (see Eq. 2). $\alpha$ is a hyperparameter that adjusts the weight of the two HMMs. Obviously, $P(\overrightarrow{A}|X)$ and $P(\overleftarrow{A}|X)$ would have opposite biases. In Figure 3c, the L2R HMM favors the path $[0, 4, 5]$ while the R2L HMM favors $[0, 1, 5]$. However, the shared parameters, namely the Transformer backbone and the emission scores $E$, force them to regularize each other during training and thus reduce their own biases implicitly. We can observe in Figure 3c that the green and red paths have equal probability now. During inference, we apply the same decoding strategy to the L2R and R2L HMMs, obtaining two translations together with their scores (i.e., probabilities). Then we only keep the translation with higher scores.

Both AW-HMM and Bi-HMM can alleviate the label bias problem, by imposing an inductive bias that tokens should be distributed evenly across all positions, making full use of the model's expressive power.

**Glancing Training**   DAT adopts the glancing strategy for training (Qian et al., 2021), which dynamically determines the number of ground truth tokens $y_t$ to be kept in the decoder input for each training instance $X$ based on current translation accuracy. For AW-HMM, the glancing strategy stays the same. As for Bi-HMM, we randomly choose the L2R or R2L model for glancing.

## 5 EXPERIMENTS

**Data**   We conducted experiments on WMT'14 En↔De, WMT'17 Zh↔En using fairseq (Ott et al., 2019). We use the preprocessed version of WMT'14 En↔De provided by fairseq[4], which contains 3.96M training sentence pairs and uses newstest2013/newstest2014 as dev/test sets. For WMT'17 Zh-En, we follow Kasai et al. (2020) for preprocessing. Note that we only use the original training data and do not apply knowledge distillation (Kim & Rush, 2016), following Huang et al. (2022c).

---

[4]http://dl.fbaipublicfiles.com/nat/original_dataset.zip

Table 1: BLEU scores on WMT'14 En↔De and WMT'17 Zh↔En. **Bold** results denote that the performances of different decoding algorithms in AW-HMM and Bi-HMM are better than their counterparts in DAT or the opposite. ‡ denotes the AW-HMM or Bi-HMM is significantly better than DAT with $p < 0.01$ and † denotes $p < 0.05$. * means the results on our machine.

| Model | Iter | WMT14 En→De | WMT14 De→En | WMT17 Zh→En | WMT17 En→Zh |
|---|---|---|---|---|---|
| Transformer* | $m$ | 27.3 | 31.6 | 24.2 | 34.9 |
| CMLM (Ghazvininejad et al., 2019) | 10 | 24.6 | 29.4 | - | - |
| CMLMC (Huang et al., 2022d) | 10 | 26.4 | 30.9 | - | - |
| Vanilla NAT (Gu et al., 2018) | 1 | 11.8 | 16.3 | 8.7 | 18.9 |
| CTC (Libovický & Helcl, 2018) | 1 | 18.4 | 23.7 | 12.2 | 26.8 |
| AXE (Ghazvininejad et al., 2020) | 1 | 20.4 | 24.9 | - | - |
| GLAT (Qian et al., 2021) | 1 | 19.4 | 26.5 | 18.9 | 29.8 |
| OaXE (Du et al., 2021) | 1 | 22.4 | 26.8 | - | - |
| CTC + GLAT (Qian et al., 2021) | 1 | 25.0 | 29.1 | 19.9 | 30.7 |
| CTC + DSLP (Huang et al., 2022a) | 1 | 24.8 | 28.3 | - | - |
| DAT(Huang et al., 2022c)* + Greedy | 1 | 25.8 | 29.6 | **22.4** | 32.5 |
|     + Lookahead | 1 | 26.3 | 29.9 | **22.7** | 33.2 |
|     + BeamSearch | 1 | 26.9 | 30.6 | 24.0 | 33.8 |
|     + Viterbi | 1 | 26.6 | 30.0 | **23.0** | 32.4 |
| AW-HMM ($\beta = 4$) + Greedy (ours) | 1 | **26.1** | **30.5**‡ | 22.3 | **33.2**‡ |
|     + Lookahead | 1 | **26.7** | **30.7**‡ | 22.4 | **33.9**‡ |
|     + BeamSearch | 1 | **27.2** | **31.2**‡ | **24.1** | **34.3**† |
|     + Viterbi | 1 | **26.8** | **30.8**‡ | 22.7 | **32.6** |
| Bi-HMM ($\alpha = 0.5$) + Greedy (ours) | 1 | **25.9** | **30.7**‡ | 22.3 | **33.9**‡ |
|     + Lookahead | 1 | **26.4** | **31.0**‡ | 22.4 | **33.9**‡ |
|     + BeamSearch | 1 | 26.7 | **31.4**‡ | 23.5 | **34.2**† |
|     + Viterbi | 1 | 26.5 | **30.7**‡ | 22.9 | **33.1**‡ |

**Model** Our implementation is based on DAT.[5] The backbone of the model is a standard Transformer, except that we remove the causal mask in decoder self-attention. The up-scale factor $\lambda$ is set to 8, following DAT (Huang et al., 2022c). The adaptive window factor $\beta = 4$ in our experiments is finetuned on the validation set of WMT'14 En-De. In Bi-HMM, the L2R/R2L weight $\alpha$ is set to 0.5, since the two models should be treated equally. We train all the models including autoregressive Transformer (Vaswani et al., 2017) for 300K steps with a batch size of 64K tokens. Other training details can be found in Appendix A. We also select the best 5 checkpoints according to the BLEU score on the validation set and average them before testing. After training, we apply greedy, lookahead, beam search (Huang et al., 2022c), and Viterbi (Shao et al., 2022) to DAT and our models. We report tokenized BLEU score (Papineni et al., 2002) for WMT'14 En↔De and WMT'17 Zh→En. For WMT'17 En→Zh, we report sacreBLEU (Post, 2018). We also report BLEURT scores (Sellam et al., 2020), a trained metric that correlates with human evaluations better than BERTScores Zhang* et al. (2020) and BLEU, in Table 2.

## 5.1 MAIN RESULTS

The main results are shown in Table 1, where all the other baselines were trained with raw data (i.e., no knowledge distillation). It can be seen that our method with different decoding algorithms can achieve comparable or better performance than DAT on all of the datasets. Specifically, on De→En and En→Zh, the lookahead performances of AW-HMM and Bi-HMM are even better than the beam search results of DAT. This may suggest that alleviating the label bias problem could help improve performance. Moreover, the beam search performances of AW-HMM and Bi-HMM are close to the autoregressive Transformer, with an average gap of 0.3 and 0.55 BLEU, while the gap between Transformer and DAT is larger, i.e., 0.7 BLEU.

---

[5] https://github.com/thu-coai/DA-Transformer

Table 2: BLEURT scores of DAT / AW-HMM / Bi-HMM. **Bold** numbers are the best results. Average scores are averaged over four decoding algorithms.

| Decoding | BLEURT (DAT / AW-HMM / Bi-HMM) | | | |
| --- | --- | --- | --- | --- |
| | De-En | En-De | En-Zh | Zh-En |
| Greedy | 62.2 / **63.5** / **63.5** | 53.4 / 53.5 / **54.5** | 57.1 / 57.3 / **57.5** | 55.5 / 55.5 / **55.9** |
| Lookahead | 62.1 / **63.5** / **63.5** | 53.0 / 53.2 / **54.3** | 56.7 / 56.9 / **57.1** | 55.1 / 55.2 / **55.7** |
| Viterbi | 62.0 / 63.1 / **63.3** | 53.3 / 53.2 / **54.6** | 55.5 / 55.6 / **56.0** | 54.0 / **54.2** / **54.2** |
| BeamSearch | 62.7 / 63.7 / **63.9** | 54.2 / 53.9 / **55.4** | 57.3 / **57.4** / 57.4 | 55.4 / **55.8** / 55.8 |
| **Average** | 62.3 / 63.5 / **63.6** | 53.5 / 53.5 / **54.7** | 56.7 / 56.8 / **57.0** | 55.0 / 55.2 / **55.4** |

Table 2 provides a comprehensive comparison of the DAT, AW-HMM, and Bi-HMM models' performance using the BLEURT metric, which is designed specifically for evaluating natural language generation tasks such as translation. Unlike the BLEU metric, BLEURT takes into account the semantic meaning and fluency of the translations, offering a more reliable assessment of translation quality. Across all decoding algorithms and language pairs, the AW-HMM and Bi-HMM models consistently achieve better or equal BLEURT scores compared to the DAT model. This consistency suggests that these models produce translations with superior semantic meaning and fluency.

One thing to note is that the performance improvements our methods can bring to Directed Acyclic Transformer (DAT) across different translation directions are closely correlated with the performance gap between DAT and the Autoregressive (AR) Transformer. As confirmed both theoretically and empirically in Sun & Yang (2020), Non-Autoregressive (NAR) models are not as expressive as AR models. Hence, we can consider the performance of the AR Transformer as a theoretical upper bound for all NAR models. According to Table 1, the DAT model can perform similarly to AR Transformer on En-De and Zh-En, with only a 0.4 and 0.2 BLEU gap. However, for De-En and En-Zh, the BLEU score differences are 1.0 and 1.1, respectively. Given that our AW-HMM/Bi-HMM are also NAR models and DAT is already pretty close to the upper bound (AR Transformer) on En-De and Zh-En, the performance gains improvements appear to be relatively modest.

## 5.2 THE EFFECT OF WINDOW SIZE $\beta$

We study the influence of adaptive window size $\beta$ on WMT'14 En-De validation set. We apply lookahead and the Viterbi algorithm to get the BLEU scores. The results are shown in Figure 4. When $\beta = 8$, this is the original setting of DAT (Huang et al., 2022c). It can be seen that the validation BLEU score first increases as the window size gets larger and then decreases.

We conjecture that there is a trade-off between combating the influence of label bias and the expressiveness of a larger window size. When the window size is small, more positions will have an equal number of outgoing transitions and thus less label bias. However, a small window size $\beta$ leads to a small set $\Gamma$, which contains less valid paths $A$ that can yield the ground truth translation $Y$ and thus less expressiveness. When the window size increases, the opposite is true. In general, $\beta = 4$ strikes a good balance between expressiveness and combating label bias. Another interesting observation is that $\beta = 7$ is slightly better than $\beta = 8$, which we attribute to the advantage of less label bias of $\beta = 7$. Since the source sentence $X$ and target sentence $Y$ have roughly the same length[6], the set $\Gamma_{\beta=7}$ should

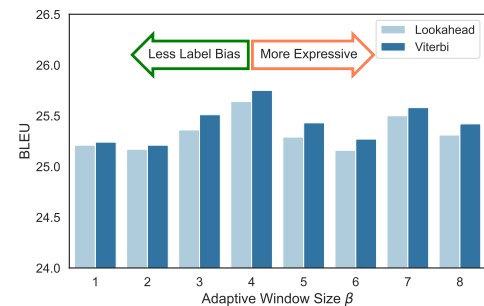

Figure 4: BLEU scores on WMT'14 En-De dev set with different adaptive window sizes $\beta$.

have a similar size to $\Gamma_{\beta=8}$. This is because $\beta = 7$ essentially prunes away many invalid transitions, which can only form invalid paths $A$ with $|A| < |Y|$. Since $\beta = 7$ has less label bias and equal expressiveness to $\beta = 8$, the performance of $\beta = 7$ should be better. More in Appendix B.

---

[6]Length ratio $|X|/|Y|$=1.06 on WMT'14 En-De dev set.

## 5.3 THE EFFECT OF L2R/R2L WEIGHT $\alpha$

Intuitively, the L2R and R2L HMMs in Bi-HMM should be treated equally, and thus the weight $\alpha$ should be 0.5. However, it is still interesting to explore how $\alpha \neq 0.5$ performs. Besides, we also compare the random glancing strategy introduced in Section 4 to an adaptive glancing strategy, which first compares $\max_{\overrightarrow{A}} P(Y, \overrightarrow{A}|X)$ and $\max_{\overleftarrow{A}} P(Y, \overleftarrow{A}|X)$ and determines the direction with a higher probability for following glancing steps.

We show the influence of different $\alpha$ on WMT'14 En-De validation set using Viterbi decoding, as shown in Figure 5. $\alpha = 1$ is the original DAT while $\alpha = 0$ is the one with opposite paths. It can be observed that $\alpha = 0.5$ is still better than other choices and random glancing performs better than adaptive glancing. We also plotted the validation BLEU score of Bi-HMM($\alpha = 0.5$) during training in Figure 6, from which we can see that Bi-HMM is more stable and can achieve better validation BLEU.

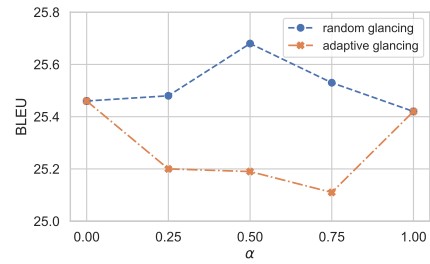

Figure 5: BLEU of models trained with random glancing/adaptive glancing on WMT'14 En-De validation set with different $\alpha$.

## 6 ANALYSIS

**Combining AW-HMM and Bi-HMM** We attempted to combine AW-HMM and Bi-HMM, i.e., imposing the window size constraint on both L2R and R2L HMMs. As shown in Table 3, we did not see an improvement in performance compared to using each approach separately. We believe this is due to the following reasons: 1) in AW-HMM, as discussed in Section 5.2 (see Figure 4), there exists a delicate balance between mitigating label bias and preserving the expressiveness offered by a larger window size in AW-HMM; 2) in the context of Bi-HMM, L2R and R2L HMMs attempt to counteract the label bias by sharing all the parameters except for the transition matrix (which contains only 0.5M paramters). Consequently, the expressive capabilities of both L2R and R2L HMMs are somewhat restrained; 3) if we impose the window size constraint of AW-HMM on both L2R and R2L HMMs, the expressiveness of both HMMs is further reduced, which consequently did not lead to improved performance.

**Measuring Label Bias** According to Section 3, label bias leads to imbalanced positioning of the generated tokens in the case of left-to-right HMM. In the toy example in Fig. 3a, the model with label bias tends to place the tokens $y_0, y_1, y_2$ at positions $0, 4, 5$ while the most balanced way of positioning should be $0, 3, 5$. Similarly, for longer sequences, we expect the model to place the tokens evenly across all positions. Hence, we can measure the influence of label bias by computing the standard deviation (STD) of the intervals between adjacent tokens:

Table 3: BLEU scores of AW-HMM, Bi-HMM and the combination of AW-/Bi-HMM (comb).

| Model | WMT'14 En-De | | |
|---|---|---|---|
| | **Greedy** | **Lookahead** | **Viterbi** |
| AW-HMM | 26.1 | 26.7 | 26.8 |
| Bi-HMM | 25.9 | 26.4 | 26.5 |
| Comb | 25.8 | 26.3 | 26.5 |

$$\sigma(\hat{A}) = \text{STD}\Big(\big[a_1 - a_0, ..., a_{|\hat{A}|-1} - a_{|\hat{A}|-2}\big]\Big)$$

Here, $\hat{A}$ is the resulting path of Viterbi decoding $\hat{Y}, \hat{A} = \arg\max_{\tilde{Y}, \tilde{A}} P(\tilde{Y}, \tilde{A}|X)$ and $a_i$ is still the position index of the generated token $\hat{y}_i$. We compute $\sigma$ for different models on the WMT'14 En-De test set, as shown in Figure 7.

We can see that the AW-HMM and Bi-HMM indeed lead to a smaller $\sigma$ than left-to-right HMM (i.e., DAT), which implies a more uniform distribution of the tokens $Y$ among all the $L$ positions and thus an indicator for less label bias. Moreover, $\sigma$ of AW-HMM seems to be quite stable across different lengths while that of Bi-HMM decreases as sentences grow longer. Besides, Bi-HMM has a smaller $\sigma$ than AW-HMM on long sentences.

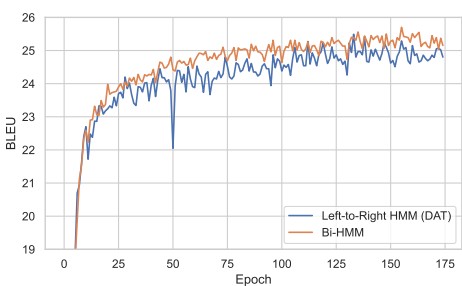

Figure 6: Validation BLEU score on WMT'14 En-De.

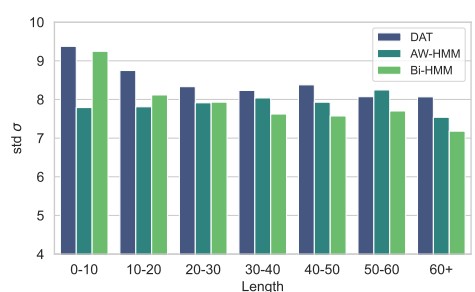

Figure 7: Standard deviation of the intervals between adjacent tokens. We group the sentences according to their lengths.

Table 4: Decoding speed of different models, with batch size 1, following previous work.

| Model | Decode | Speed |
|---|---|---|
| Transformer | - | 1.0× |
| DAT / AW-HMM / Bi-HMM | Greedy | 12.2× / 12.3× / 12.0× |
| | Lookahead | 12.2× / 11.8× / 11.5× |
| | Viterbi | 10.8× / 10.9× /  8.0× |

**Decoding Speed**   We also evaluated the decoding speed of various models on a single RTX-6000 GPU, as shown in Table 4. The AW-HMM model exhibits a decoding speed comparable to that of the DAT model, while the Bi-HMM model has a marginally slower Viterbi decoding speed. The reduced speed for the Bi-HMM model can be attributed to the need to decode both the Left-to-Right (L2R) and Right-to-Left (R2L) parts. However, it is important to note that, in theory, the L2R and R2L decoding processes can be executed simultaneously, which could potentially compensate for the speed loss experienced in the Bi-HMM model.

**L2R/R2L Direction Preferences**   In this section, we study how the L2R/R2L parts in Bi-HMM perform and the BLEU upper bound (UB) of Bi-HMM, as shown in Table 5.   For **L2R/R2L**, we simply decode the respective L2R/R2L parts in Bi-HMM. For **UB**, we compare the sentence BLEU scores of the translations generated L2R and R2L parts with respect to the references, and only keep the one with higher BLEU. Then, we compute the corpus BLEU of the filtered translations. First, we can observe that Bi-HMM does not have a consistent preference over the L2R or R2L direction, which suggests that L2R/R2L directions are equally expressive. Next, the UB BLEU is much higher than the Bi/L2R/R2L performance, which suggests that there might be some potential opportunities for further improvements, e.g., how to decode the bi-directional model other than referring to the log probability.

Table 5: BLEU scores of Bi-HMM (Bi), L2R part, R2L part, upper bound (UB) on WMT'14 En-De test set.

| Decoding | WMT'14 En-De | | | |
|---|---|---|---|---|
| | **Bi** | **L2R** | **R2L** | **UB** |
| Lookahead | 26.4 | 26.2 | 25.7 | **28.7** |
| Viterbi | 26.7 | 26.6 | 25.9 | **29.0** |
| BeamSearch | 26.7 | 26.7 | 26.7 | **29.1** |

# 7   RELATED WORK

**Non-Autoregressive Machine Translation**   Gu et al. (2018) proposed non-autoregressive (NAR) machine translation to accelerate inference. To facilitate training, sequence-level knowledge distillation (Kim & Rush, 2016) is often applied. Various latent variable models were developed to enhance the NAR model's ability to handle multi-modality problem (Ma et al., 2019; Lee et al., 2018; 2020; Saharia et al., 2020; Shu et al., 2020; Gu & Kong, 2021; Bao et al., 2022). Researchers also proposed various loss functions and training strategies to facilitate model learning (Libovický

& Helcl, 2018; Shao et al., 2020; Ghazvininejad et al., 2020; Du et al., 2021; Qian et al., 2021; Zhan et al., 2022; Huang et al., 2022a;d; Qian et al., 2022; Ma et al., 2023). More recently, Huang et al. (2022b) proposed a unified framework for non-autoregressive Transformer learning. We refer readers to Xiao et al. (2022) for a more complete review of NAT.

**Label Bias**    Label bias was first studied in Bottou & Fogelman-Soulié (1991) and it is also the motivation for developing CRF (Lafferty et al., 2001). Label bias was also studied in neural-based models (Andor et al., 2016; Goyal, 2022), where it is closely related to the local normalization. Goyal et al. (2022) studied the sequence-level sampling method for non-autoregressive models like BERT (Kenton & Toutanova, 2019) by exposing the implicit globally normalized energy networks. Developing efficient globally normalized models for both AR and NAR models in machine translation remains an active research area.

## 8    CONCLUSION

In this paper, we argue that directed acyclic Transformer (Huang et al., 2022c) is essentially a left-to-right HMM, with the source and target sequences $X, Y$ being observations and the token positions $A$ being latent states. Furthermore, this left-to-right may suffer from the label bias issue due to the unavailability of the target observations $Y$ during inference, which could under-utilize the model's expressive power. Therefore, we propose Adaptive Window HMM (AW-HMM) and Bi-directional HMM (Bi-HMM) to combat label bias. The experimental results on WMT'14 En↔De and WMT'17 En↔Zh show the effectiveness of our methods and we also quantitatively show that our method indeed mitigates the influence of label bias.

## REPRODUCIBILITY

We provide the source code in the supplementary materials. We also provide the hyperparameters and the downloadable training data in Section 5 and Appendix C. We'd like to provide any details regarding the experiments upon request.

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

## A EXPERIMENT DETAILS

Our experiment settings follow Huang et al. (2022c). We use Transformer (base) (Vaswani et al., 2017) as our backbone. We set dropout to 0.1, weight decay to 0.01 and label smoothing 0.1 for Transformer. We train all models for 300K steps with a batch size of 64K tokens. Our learning rate warms up to $5 * 10^{-4}$ in the first 10K steps and then decays with the inverse square-root schedule. Since our implementation is not based on CUDA, the training cost of AW-HMM/Bi-HMM is 46/56 hours on 8 A100 GPUs. The training cost of CUDA-optimized DAT is 42 hours on 8 A100 GPUs.

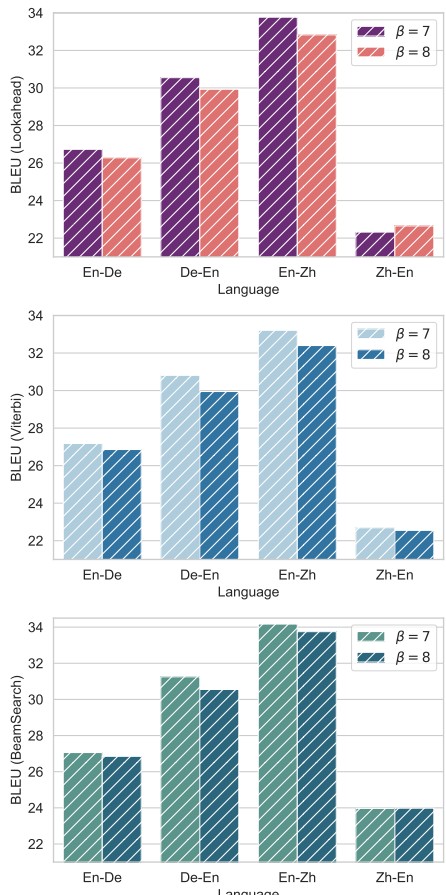

Figure 8: The test BLEU of AW-HMM with $\beta = 7, 8$ on WMT'14 En↔De, WMT'17 En↔Zh.

## B  WINDOW SIZE $\beta = 7$ VS. $\beta = 8$

Here, we list the test results of AW-HMM with $\beta = 7, 8$ in Figure 8. It can be seem that $\beta = 7$ is slightly better than $\beta = 8$ on most benchmarks.

## C  GENERATION DIVERSITY

To test the generation diversity of DAT as well as our AW-HMM and Bi-HMM, we sample from the transition and emission probabilities in Eq. 1, using nucleus sampling (Holtzman et al., 2020) with $p = 0.8$ and varying temperatures. Then we compute the Pairwise-BLEU and multi-reference BLEU (Shen et al., 2019) on a multi-reference set provided by Ott et al. (2018), which contains 10 additional human-written references for each of the 500 sentences from WMT'14 En-De test set. Lower Pairwise-BLEU and higher multi-reference BLEU indicate more diversity and higher quality of the translations. As shown in Figure 9, AW-HMM is more close to human performance than DAT, and does not sacrifice its generation diversity due to its smaller window size ($\beta = 4$). Bi-HMM, however, is more diversified than DAT when the temperature is low, i.e., 0.4, 0.6. Besides, the L2R and R2L parts have different behaviors, with the L2R part exhibiting less diversity and the R2L part being more diverse.

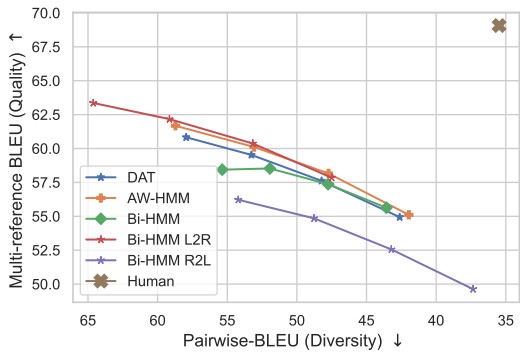

Figure 9: Pairwise-BLEU and Multi-reference BLEU (leave-one-out) of different models (Shen et al., 2019). We sample with temperatures $[0.4, 0.6, 0.8, 1.0]$ (from left to right). We sample 10 hypotheses for each instance.

## D   PARAMETER COUNT

We list our model parameters in Table 6. The table indicates that our AW-HMM/Bi-HMM and DAT models only require a $3 - 5\%$ increase in parameter count compared to the baseline models. This suggests that our performance improvements are due primarily to our algorithm, rather than an increase in parameters.

|  | En-De | En-Zh |
| --- | --- | --- |
| Transformer | 65M | 86M |
| DAT | 67M | 88M |
| AW-HMM | 67M | 89M |
| Bi-HMM | 68M | 89M |

Table 6: The parameter count of different models.

