# OpenReview forum: "Non-Autoregressive Machine Translation as Constrained HMM"
_ICLR.cc/2024/Conference — Submitted to ICLR 2024_

### Official Review · Reviewer_z6kj · 2023-10-27

**Soundness:** 3 good
**Presentation:** 3 good
**Contribution:** 3 good
**Rating:** 8
**Confidence:** 4

**Summary:**

This paper proposes that DAT can be considered a special case of HMM, and then utilizes this perspective to identify that DAT exits the label bias problem. To address this problem, the authors present two solutions, namely 1) adaptive window HMM and 2) bi-directional HMM. Experimental results on WMT'14 English to German and WMT'17 Chinese to English demonstrate that our methods can achieve better or comparable performance to the original DAT.

**Strengths:**

1) Viewing DAT as a variant of HMM is correct and very helpful. As a broader and high-level perspective, HMM can provide more opportunities for improving NAT (DAT).
2) Label bias is indeed an issue with DAT, and the two solutions proposed by the authors are simple but effective. The intuition behind them is also easy to understand.
3) The experiments are very thorough, verifying not only the improvement in performance but also analyzing whether the label bias issue has been resolved in the analysis section.
4) This paper is clear and easy-to-follow.

**Weaknesses:**

I cannot point obvious shortcomings, but if pressed, I would argue that label bias is not the most critical issue within DAT. In other words, this paper is not a game changer for NAT. From the experiments, it appears that addressing label bias offers only limited enhancement to DAT's performance. However, this cannot be considered a very strong point of criticism, as I think the authors' perspective of viewing DAT through the lens of HMM to be very useful and improtant.

**Questions:**

I've also entertained the idea of viewing DAT as an HMM and have conducted some preliminary experiments. For instance, I removed the lower triangular mask matrix in DAT, transforming the model into a globally normalized general HMM. However, the model did not converge. If you could contrast this unsuccessful method in your paper, we might gain a deeper understanding of HMM-DAT.

Additionally, in DAT experiments, glancing training significantly aids DAT. Do you think this training method can be generalized to all HMM algorithms, such as those used in speech recognition, etc.?

---

> ### Author Response · Authors · 2023-11-21
>
> We sincerely appreciate your thorough and thoughtful review and your recognition of the significance of our work. It's truly encouraging to know that our efforts to view DAT as a variant of HMM and address the label bias issue have been understood and valued.
>
> ### **About the Lower Triangular Mask in DAT**
> We understand that you attempt to treat DAT as a conventional HMM, where the transition probability between any two states can be greater than zero, instead of a left-to-right HMM where we only allow left-to-right transitions.
>
> To illustrate the importance of the lower triangular mask (LTM) for optimizting the objective function in DAT, let's start with a toy example. This example is similar to but slightly more complex than the one in Figure 3(a) of our paper.
> Assume we have 6 positions (or latent states, 0-5) and 4 target tokens `[BOS, y1, y2, EOS]`. We force the start and end of the target sequence to be the first and the last positions, i.e., 0 and 5. Thus, position 0 and 5 are responsible for predicting the special BOS and EOS tokens.
>
> **Removing the mask confuses training**
>
> With the LTM, the latent paths that can yield the target sequence `[BOS, y1, y2, EOS]` are `[0,1,2,5], [0,1,3,5], [0,1,4,5], [0,2,3,5], [0,2,4,5], [0,3,4,5]`.
> However, without the LTM, we would have additional latent paths, namely, `[0,2,1,5], [0,3,1,5], [0,4,1,5], [0,3,2,5], [0,4,2,5],[0,4,3,5]`. Note that an additional path like `[0,2,1,5]` corresponds to tokens `[BOS, y1, y2, EOS]`, respectively.
>
> According to the Eq. (5) in the original DAT paper [1], only one latent path will stand out during the training process for each training instance, and this standing-out path can be any valid path. For example, with LTM, `[0,1,2,5]` might be the most probable path for the target sentence `[BOS, the, world, EOS]` while `[0,1,3,5]` is the most likely one for `[BOS, the, earth, EOS]`.
> Without LTM, `[0,1,2,5]` might still be chosen for `[BOS, the, world, EOS]`, but `[0,2,1,5]`, which has a right-to-left transition 2->1, might be selected for `[BOS, the, earth, EOS]`.
> Thus, when we keep the LTM, the model is more likely to share some learned patterns (e.g., placing "the" at position 1) between different training examples.
> Nevertheless, when we remove the LTM, half of the training instances will likely to have a winning left-to-right latent path whereas another half choose right-to-left. This is because the DAT model parameters are randomly initialized at the start of training. Consequenttly, this random L2R-R2L choices will make it hard for the model to share learned patterns among the training examples. Hence, the model cannot compress the training data effectively and struggles to converge.
>
>
> **Removing the mask creates loops during inference**
>
> Another reason for keeping the LTM is that we can avoid loops in the decoding path, which is important for generating a complete sequence. For example, if we remove the LTM, and we happen to have the transition probability `P(1->2)=0.99` and `P(2->1)=0.99` during greedy inference, arriving at position 1 would make position 2 the most likely next position. However, once at position 2, the most likely candidate becomes position 1, which leads to a loop. We will never reach the EOS token (the last position) and simply generate a repetitive and meaningless sequence.
>
> This LTM is important for both training and inference, and we sincerely thank you for asking this question so that more readers can understand.
>
> ### **About Glancing Training in other HMMs**
>
> This is an interesting question! Even though we are not experts in the speech domain, we'd like to share our understanding of glancing training in HMM-DAT.
>
> Ideally, the DAT decoder inputs should be only the learnable positional embeddings plus the `unk` token embeddings, but with glancing training, we replace some `unk` tokens with the ground truth target tokens, which provides richer training signals in the forward pass as well as rich gradient info in the backward pass, and thus leads to better convergence. By richer, we refer to the fact that different training examples will provide different glancing input tokens with glancing training. Without glancing, the inputs are always the same, consisting only of the unk token plus positional embedding.
> If you are using a non-autoregressive decoder in speech recognition, you can similarly adopt the glancing training strategy.
> We are happy to discuss more on this topic if you don't mind providing more background information or related papers.
>
> Thank you once again for your insightful review. We hope our response has addressed your queries satisfactorily. We look forward to further discussions and exchanges our views.
>
>
> [1]: Directed Acyclic Transformer for Non-Autoregressive Machine Translation (Huang Fei et al., ICML 2022)

---

> > ### Comment · Reviewer_z6kj · 2023-11-22
> >
> > Thanks for your detailed reply! Your paper is really good.

---

> > > ### Author Response · Authors · 2023-11-22
> > >
> > > Thank you very much for your kind words and positive feedback on my paper!

---

### Official Review · Reviewer_bRfo · 2023-10-30

**Soundness:** 3 good
**Presentation:** 3 good
**Contribution:** 2 fair
**Rating:** 5
**Confidence:** 4

**Summary:**

Based on the directed acyclic Transformers (DAT) for non-autoregressive translation (NAT), this paper first shows that NAT is a fully connected left-to-right Hidden Markov Model (HMM) model. Then, the authors propose two constrained HMM strategies to address label bias issues in DAT, including adaptive window HMM and bidirectional HMM. The former adaptively balances the number of outgoing transitions at different latent states. And the latter uses bidirectional components to regularize each other's label bias.

The experiments are conducted on WMT14 en-de and WMT17 zh-en. Results demonstrate that both proposed strategies can obtain comparable or better performance compared to previous DAT models, and reduce the influence label bias.

**Strengths:**

1. The paper proposes two methods, namely adaptive window HMM and bidirectional HMM to alleviate the challenges of label bias.

2. Experimental results and analysis demonstrate the effectiveness of the proposed methods, which can achieve comparable or better BLUE scores than the original DAT models, and mitigate the effect of label bias.

**Weaknesses:**

Compared to original DAT methods, the proposed strategies are incremental innovation, and only achieve improvements on the part of translation directions. For example, it does not seem to work for WMT zh-en, the reasons also need explaining.

**Questions:**

1. Can the proposed two strategies be applied to the DAT model at the same time?

2. Why does the proposed method behave differently in different translation directions?

---

> ### Author Response · Authors · 2023-11-21
>
> Thank you for your insightful review and constructive feedback. We appreciate your time and effort, which has helped us improve our paper.
>
> **Response to Question 1**
>
> We attempted to integrate two strategies, but we did not see an improvement in performance compared to using each approach separately. We believe this is due to the following reasons:
> - In AW-HMM, as discussed in Section 5.2 (see Figure 4), there exists a delicate balance between mitigating label bias and preserving the expressiveness offered by a larger window size in AW-HMM.
> - In the context of Bi-HMM, L2R and R2L HMMs attempt to counteract the label bias by sharing all the parameters except for the transition matrix (which contains only 0.5M paramters). Consequently, the expressive capabilities of both L2R and R2L HMMs are somewhat restrained.
> - If we impose the window size constraint of AW-HMM on both L2R and R2L HMMs, the expressiveness of both HMMs is further reduced, which consequently did not lead to improved performance.
>
> We appreciate this insightful question and have incorporated a detailed discussion on this phenomenon in our revised manuscript (see `the first paragraph in Section 6`).
>
> **Response to Question 2**
>
> The performance improvements our methods can bring to Directed Acyclic Transformer (DAT) across different translation directions are closely correlated with the performance gap between DAT and the Autoregressive (AR) Transformer.
> As confirmed both theoretically and empirically in [2], Non-Autoregressive (NAR) models are not as expressive as AR models.
> Hence, we can consider the performance of the AR Transformer as a theoretical upper bound for all NAR models.
> We have tabulated the best NAR model performances for all translation directions from the Table 1 of our paper, and compare them with the AR Transformer.
>
> |             | En-De | De-En | Zh-En | En-Zh |
> |-------------|-------|-------| ------ | ----- |
> | Transformer | 27.3  | 31.6 | 24.2 | 34.9 |
> | DAT         | 26.9 (-0.4)| 30.6 (-1.0)|24.0 (-0.2) | 33.8 (-1.1) |
> | AW-HMM/Bi-HMM | 27.2 (-0.1)| 31.4 (-0.2)| 24.1 (-0.1)|34.3 (-0.6)|
> |
>
> As can be seen, the DAT model can perform similarly to AR Transformer on En-De and Zh-En, with only a `0.4` and `0.2` BLEU gap. However, for De-En and En-Zh, the BLEU score differences are `1.0` and `1.1`, respectively.
> Given that our AW-HMM/Bi-HMM are also NAR models and DAT is already pretty close to the upper bound (AR Transformer) on En-De and Zh-En, the performance gains improvements appear to be relatively modest.
>
> This point has been clarified and added to our revised paper (see `the last paragraph in Section 5.1`). We sincerely thank you for bringing up this important issue.
>
>
> We hope that our response has satisfactorily addressed your questions. If there are any additional questions or points of clarification you require, we respectfully invite you to share them.
>
>
> **References**
>
> [1]: Directed Acyclic Transformer for Non-Autoregressive Machine Translation (Huang Fei et al., ICML 2022)
>
> [2]: An EM Approach to Non-autoregressive Conditional Sequence Generation (Zhiqing Sun, Yiming Yang, ICML 2020)

---

> > ### Author Response · Authors · 2023-11-23
> >
> > Dear Reviewer bRfo,
> >
> > Hope you're well. We wanted to follow up on the responses submitted during the rebuttal period. Given that we are just 8 hours away from the deadline, we were hoping to get any additional feedback or comments you might have.
> >
> > We understand everyone's schedules are busy, and we appreciate the time and effort you've already dedicated to reviewing our work. Your insights are extremely valuable.
> >
> > Thank you in advance for your time.

---

### Official Review · Reviewer_fXmy · 2023-10-30

**Soundness:** 2 fair
**Presentation:** 2 fair
**Contribution:** 2 fair
**Rating:** 3
**Confidence:** 4

**Summary:**

The presented work extends the understanding of DAT as a left-ro-right Hidden Markov Model and proposes two approaches to mitigate the inherent label bias problem, namely an Adaptive Window HMM and a combination with a right-to-left HMM.

**Strengths:**

- Extends the understanding of DAT as a HMM and solves the label bias problem by incorporating an R2L HMM and adding a hyper parameter to balance the outgoing transitions.
- Experiments to back up the claim that the label bias problem is mitigated using the proposed approach.
- NAT papers should follow the broader machine translation standard to report multiple metrics and metrics that correlate better with human judgement besides only relying on tokenized BLEU as that doesn't show the full picture, see **[1]**, **[2]**, **[3]**. I'm glad to see that BLEURT was additionally reported in the presented work and we do see nice gains there as well.

**Weaknesses:**

### Weaknesses

- **[major]**: Despite WMT'14 and WMT'17 being commonly used in the NAT literature, they are now way overhauled in the broader machine translation literature and should be replaced by more recent test sets to put the results into the context of recent research, see **[1]**.
- **[major]**: NAT papers should follow the broader machine translation standard to report their evaluation scores using `sacrebleu` and provide the corresponding hash that was used for generating the scores. This will ensure that scores are reproducible and do not vary across papers by up to 1.8 BLEU points due to varying tokenization and normalization, see **[2]**, **[4]**. Mixing `sacrebleu` and tokenized BLEU as done in Table 1 shouldn't be done and needs to be fixed.
- **[major]**: While it is nice to see that the paper attempts to provide GPU benchmarking numbers, the speed multipliers are heavily inflated since the baseline is a non-optimized autoregressive Transformer. There are many de-facto standard ways in practice to construct are more competitive autoregressive inference speed baseline with negligible translation quality drop using e.g. shallow decoders, shortlisting, or quantization (see **[2]**, **[5]**, **[6]**) which should be adopted here.
- **[major]**: Table 1 doesn't include parameter counts or inference speed numbers which makes it hard to compare the different approaches and understand if the improvement comes from the better algorithm or, simply, the increased parameter count capacity. For example, Bi-HMM uses two different parameter sets to model L2R and R2L and as a result they should have more parameters. Bigger baselines, potentially in parallel branches through e.g. MoE, or scaling up previous approaches might be needed.
- **[minor]**: It is unclear how well the proposed approach extends to the multilingual setting.
- **[minor]**: Figure 4 doesn't show a clear trend in the window size, making it hard to extrapolate the findings to other language pairs or datasets without additional analysis for the dataset at hand. This will require additional grid search tuning trials to adopt and no guidance on how to tune this parameter is given.

---
### References

- **[1]**: [Non-Autoregressive Machine Translation: It’s Not as Fast as it Seems](https://aclanthology.org/2022.naacl-main.129) (Helcl et al., NAACL 2022)
- **[2]**: [Non-Autoregressive Neural Machine Translation: A Call for Clarity](https://aclanthology.org/2022.emnlp-main.179) (Schmidt et al., EMNLP 2022)
- **[3]**: [Results of WMT22 Metrics Shared Task: Stop Using BLEU – Neural Metrics Are Better and More Robust](https://aclanthology.org/2022.wmt-1.2) (Freitag et al., WMT 2022)
- **[4]**: [A Call for Clarity in Reporting BLEU Scores](https://aclanthology.org/W18-6319) (Post, WMT 2018)
- **[5]**: [Findings of the WMT 2022 Shared Task on Efficient Translation](https://aclanthology.org/2022.wmt-1.4) (Heafield et al., WMT 2022)
- **[6]**: [Edinburgh’s Submission to the WMT 2022 Efficiency Task](https://aclanthology.org/2022.wmt-1.63) (Bogoychev et al., WMT 2022)

**Questions:**

- How were the hyperparameters tuned for the proposed method and the previous works? If defaults were used for previous methods, the comparison needs to potentially be adjusted to also allow hyper parameter tuning for those methods.

---

> ### Comment · Reviewer_z6kj · 2023-11-11
> **Strong Disagreement**
>
> In my view, the first two weaknesses you mentioned are based on a bias against the NAT field rather than inherent issues with the paper itself. When evaluating a paper, we should focus more on the content of the paper itself, rather than imposing the problems of the entire field on a single paper. Moreover, I believe the issues with BLEU are not intentionally caused by the authors, as previous works have used BLEU for EN-DE and DE-EN, so it is acceptable to use BLEU for fairness. The authors also reported SACREBLEU results for WMT 17 EN-ZH.
>
> I think this paper has a deep understanding of DAT (in the view of HMM). Beyond the  acceleration advantages of NAT this model is also a variant different from the existing autoregressive models. I believe the academic community must encourage such exploration, otherwise, neural machine translation (NMT) will never replace SMT. To my knowledge, the performance of early NMT models (e.g., RNN Search) was not significantly better than SMT.

---

> > ### Comment · Reviewer_fXmy · 2023-11-11
> >
> > > When evaluating a paper, we should focus more on the content of the paper itself, rather than imposing the problems of the entire field on a single paper.
> >
> > There are multiple works now that mention the above points as a big problem in the NAT literature and inherently issues within a subfield fall back to the individual papers. If obvious issues in the evaluation are not used as gating criteria for paper acceptance, the subfield will never change and remain its problematic state. Frankly, it is not much effort to actually have a sound evaluation procedure and multiple papers have now demonstrated how to do it properly. I firmly stand by my points and would like to see a revised version of the paper.
> >
> > > Moreover, I believe the issues with BLEU are not intentionally caused by the authors, as previous works have used BLEU for EN-DE and DE-EN, so it is acceptable to use BLEU for fairness. The authors also reported SACREBLEU results for WMT 17 EN-ZH
> >
> > I do acknowledge in my previous review that BLEURT was additionally used and I'm quite happy to see that. Still, the issue I mentioned in my original review stands where mixing scores from `sacrebleu` and tokenized BLEU is confusing and shouldn't be done. Again, it's not much effort to switch to a sound evaluation procedure and multiple papers have demonstrated existing issues with the current approach.
> >
> > > I believe the academic community must encourage such exploration
> >
> > I agree and am very much in favour for encouraging progress in this subfield. However, common best practices from the broader machine translation community should not be disregarded to do so such that meaningful progress can actually be identified. In its current state, neither the translation quality nor the inference speedup are properly evaluated.

---

> ### Author Response · Authors · 2023-11-21
>
> Thank you for taking time to review our paper.
>
> Firstly, we would like to clarify that **we aim to improve the best NAR model from a novel theorectical perspective, not to push the translation frontier**.
>
> The primary focus of our study is on the extension of the understanding of the Directed Acyclic Transformer (DAT) as a left-to-right Hidden Markov Model (HMM) and to address the inherent **label bias** problem with our AW-HMM/Bi-HMM approaches. Our objective was to delve deeper into the theoretical aspects of this model and propose improvements in the same, rather than advancing the translation landscape. We believe our findings are original and can add new knowledge to our community.
>
> The concerns you raise about the broader Non-Autoregressive Translation (NAT) community, while relevant to the field as a whole, fall outside the scope of this particular study.
> We must highlight that even after the publication of DAT, many research papers continue to use WMT'14, WMT'16, WMT'17 translation tasks as the testbed for NAR models, e.g., `[2],[3],[4],[5],[6],[7],[8],[9],[10]`. Even the authors of `[11]`, whom you cited, used WMT'13, 14, 16 for their evaluations, as their goal was also not to push the translation frontier.
>
> As `Reviewer z6kj` accurately highlighted, the potential limitations existing within the broader NAT community should not overshadow the specific contributions made in individual studies, such as ours, which aims to enhance the state-of-the-art NAR architecture from a novel HMM perspective.
>
>
> **About BLEU and SacreBLEU**
>
> We choose to report BLEU for EN-DE and SacreBLEU for ZH-EN for the sake of a consistent and fair comparison with previous work, as it was hard to accurately reproduce all the previous baselines and report their SacreBLEU scores.
>
> **About GPU speedup numbers**
>
> Comparing a non-optimized NAR model with an optimized AR model would be unfair. The AR model field has various optimization techniques for acceleration, while the NAR field, being at its early stage, lacks specially-designed speedup techniques.
> However, it is crucial to note that the time complexity of NAR/AR model is $O(N)/O(N^2)$ [12], respectively. Theoretically, as indicated in [12], NAR models can surpass AR models in terms of decoding speed under any test conditions, provided that they are properly optimized. We look forward to such advancements in NAR accerleration, but again, it falls outside the scope of this paper.
>
>
> **About Parameter Count**
>
> We list our parameter count below:
> |             | En-De | En-Zh |
> |-------------|-------|-------|
> | Transformer | 65M   | 86M   |
> | DAT         | 67M   | 88M   |
> | AW-HMM      | 67M   | 88M   |
> | Bi-HMM      | 68M   | 89M   |
> |
>
> The table indicates that our AW-HMM/Bi-HMM and DAT models only require a `3%-5%` increase in parameter count compared to the baseline models. This suggests that our performance improvements are due primarily to our algorithm, rather than an increase in parameters.
>
> This information has been added to our revised paper (see Appendix D).
>
>
> In response to your minor queries:
>
> **Response to [minor 1]**
>
> Our models/methods are not language-specific, and should thus be applicable to other language pairs. We chose En-De and En-Zh to maintain consistency with previous work, as in [1].
>
> **Response to [minor 2]**
>
> Regarding the window size trend experiment depicted in Figure 4, we confined our investigation to the WMT'14 En-De dataset. This decision was made due to the significant computational cost associated with examining 8 different window sizes (ranging from 1 to 8) for each of the 4 translation directions (En-De, De-En, Zh-En, En-Zh).
>
> However, please note that we have conducted experiments with window sizes of 7 and 8 for all language pairs, the results of which are documented in Appendix B.
>
> In relation to the tuning of the window size $\beta$ hyperparameter, we followed standard hyperparameter tuning practices. The hyperparameter was tuned on the development set, a process that is mentioned in the caption of Figure 4.
>
> **Response to Questions**
>
> Our training hyperparamters follow previous work [1]. For the hyperparmeters in our AW-HMM and Bi-HMM, we tune them on the dev set.
>
> We welcome further questions regarding the DAT-as-an-HMM perspective and our Bi-/AW-HMM models and would be glad to elaborate on these points.

---

> ### Author Response · Authors · 2023-11-21
> **add references due to char limit**
>
> [1]: Directed Acyclic Transformer for Non-Autoregressive Machine Translation (Huang Fei et al., ICML 2022)
>
> [2]: Rephrasing the Reference for Non-autoregressive Machine Translation (Chenze Shao et al., AAAI 2023)
>
> [3]: Selective Knowledge Distillation for Non-Autoregressive Neural Machine Translation (Min Liu et al, AAAI 2023)
>
> [4]: Diff-Glat: Diffusion Glancing Transformer for Parallel Sequence to Sequence Learning (Lihua Qian et al., Dec 2022)
>
> [5]: DePA: Improving Non-autoregressive Machine Translation with Dependency-Aware Decoder (Jiaao Zhan et al., IWSLT 2023)
>
> [6]: Deep Equilibrium Non-Autoregressive Sequence Learning (Zaixiang Zheng et al., ACL 2023)
>
> [7]: Viterbi Decoding of Directed Acyclic Transformer for Non-Autoregressive Machine Translation (Chenze Shao et al., EMNLP 2022)
>
> [8]: Non-autoregressive Streaming Transformer for Simultaneous Translation (Zhengrui Ma et al., Oct 2023)
>
> [9]: DINOISER: Diffused Conditional Sequence Learning by Manipulating Noises (Jiasheng Ye et al., 2023)
>
> [10]: AMOM: Adaptive Masking over Masking for Conditional Masked Language Model (Yisheng Xiao et al., AAAI 2023)
>
> [11]: Non-Autoregressive Neural Machine Translation: A Call for Clarity (Schmidt et al., EMNLP 2022)
>
> [12]: Deep Encoder, Shallow Decoder: Reevaluating Non-Autoregressive Machine Translation (Jungo Kasai et al., ICLR 2021)

---

> > ### Author Response · Authors · 2023-11-23
> >
> > Dear Reviewer fXmy,
> >
> > We hope this message finds you well. We appreciate the time and effort you've already dedicated to reviewing our work. We've submitted responses during the rebuttal period and we're eager to hear your further thoughts.
> >
> > With the deadline just 8 hours away, we understand everyone's schedule is tight. However, we would greatly appreciate if you could provide us with any additional feedback or comments.
> >
> > Thank you in advance for your time and consideration.

---

> > > ### Comment · Area_Chair_ERpm · 2023-12-05
> > >
> > > @fXmy
> > >
> > > Do you have any new comments to add after reading the author's response?

---

> > > > ### Comment · Reviewer_fXmy · 2023-12-05
> > > > **Sorry for the delay**
> > > >
> > > > > We choose to report BLEU for EN-DE and SacreBLEU for ZH-EN for the sake of a consistent and fair comparison with previous work, as it was hard to accurately reproduce all the previous baselines and report their SacreBLEU scores.
> > > >
> > > > This alone verifies the problematic state the subfield is in. I'd much rather have less previous work to compare to but a sound evaluation procedure for a subset of previous works.
> > > >
> > > > > Comparing a non-optimized NAR model with an optimized AR model would be unfair.
> > > >
> > > > Comparing NAR models to AR models without taking into account existing and de-facto standard speedup techniques for AR models paints an unfair picture and heavily overestimates the speedup. In fact, many of the speedup techniques for AR can be applied to NAR models as well (e.g. shortlisting, quantization, shallow decoders) and we should compare them on the basis what is currently possible not only based on theoretical guarantees.
> > > >
> > > > > Our models/methods are not language-specific, and should thus be applicable to other language pairs.
> > > >
> > > > Moving from any uni- / bi-directional setup to a MNMT setup comes with its own quirks in training and my initial question was aimed towards that there were no comparisons regarding that setup (see [Can Multilinguality Benefit Non-autoregressive Machine Translation?](https://openreview.net/forum?id=J_o_J6B1wJ-)).
> > > >
> > > > > In relation to the tuning of the window size $\beta$ hyperparameter, we followed standard hyperparameter tuning practices
> > > >
> > > > What exactly are the "standard" tuning practices for this hyperparameter? For e.g. $\eta$ in Adam the sampling distribution e.g. uniform / log-uniform / etc. has quite an effect on the final performance so some guidance on tuning distributions / grid search ranges / or anything along those lines would be tremendously helpful.
> > > >
> > > > ---
> > > > ---
> > > >
> > > > @AC: I still stand by my points that having a sound evaluation procedure both in terms of translation quality and inference speed (the main argument in favour of NAR methods) is a requirement for publication that this paper simply does not fulfil. I do acknowledge that **some** previous work is still able to get published while still having all of these issues as well but I, and many other researchers in the broader machine translation community, share the sentiment that the NAR subfield needs to start properly evaluating approaches before meaningful progress can be made. Unfortunately, this falls back onto the individual papers and unless we start enforcing it during the publication process, it will never change. I'd be happy to hop on a quick call with other reviewers / AC to make my point if required but in its current state I am still in favour of rejecting.

---

### Meta-Review · Area_Chair_ERpm · 2023-12-10

**Metareview:**

**Summary**

This paper studies the directed acyclic Transformers (DAT) for non-autoregressive translation (NAT) which can be seen as a left-to-right HMM. Then, two strategies (adaptive window HMM, bidirectional HMM) are proposed to migrate the inherent label bias problem existed in DAT (HMM). The results on WMT14 en-de and WMT17 zh-en show that both proposed strategies can obtain comparable or better performance compared to previous DAT models.

**Strength**

1. Formulating DAT as HMM improves the high-level understanding for designing new methods to improve non-autoregressive text generation.
2. The experimental results indicate the label bias problem is effectively eased using the proposed two approaches. The results are validated on multiple metrics to verify the main claim.

**Weaknesses**

1. Compared to the original DAT methods, the proposed method is quite incremental, and the improvement is also evident enough. This diminishes the overall contribution of accepting this paper.
2. Experiments are conducted on WMT14 En-De and WMT17 EN-ZH only. It is suggested that more comparisons be conducted with more recent test sets and in multilingual settings.
3. The paper does not include a comparison with more optimized autoregressive Transformers as baselines, such as shallow decoders, shortlisting, etc. Also, as raised by fXmy, many of the comparison details (e.g., parameter counts, inference speed) are missing, which makes it difficult to draw good conclusions.
4. The ablation study in window size lacks the necessary details to draw conclusions.
5. The proposed two methods are not straightforward to be combined.

**Justification For Why Not Higher Score:**

The current approaches are a bit incremental, and the experiments may not be enough to justify that the proposed approaches are actually much more useful than the original DAT. In this case, more experiments or analyses should be useful to confirm that.

**Justification For Why Not Lower Score:**

N/A

---

### Decision · Program_Chairs · 2024-01-16

Reject